# Visualization of the type III secretion mediated *Salmonella*–host cell interface using cryo-electron tomography

Donghyun Park[1,2], Maria Lara-Tejero[1], M Neal Waxham[3], Wenwei Li[1,2], Bo Hu[4,5], Jorge E Galán[1], Jun Liu[1,2,4]*

[1]Department of Microbial Pathogenesis, Yale University School of Medicine, New Haven, United States; [2]Microbial Sciences Institute, Yale University School of Medicine, New Haven, United States; [3]Department of Neurobiology and Anatomy, McGovern Medical School, The University of Texas Health Science Center at Houston, Texas, United States; [4]Department of Microbiology and Molecular Genetics, McGovern Medical School, The University of Texas Health Science Center at Houston, Texas, United States; [5]Department of Pathology and Laboratory Medicine, McGovern Medical School, The University of Texas Health Science Center at Houston, Texas, United States

**Abstract** Many important gram-negative bacterial pathogens use highly sophisticated type III protein secretion systems (T3SSs) to establish complex host-pathogen interactions. Bacterial-host cell contact triggers the activation of the T3SS and the subsequent insertion of a translocon pore into the target cell membrane, which serves as a conduit for the passage of effector proteins. Therefore the initial interaction between T3SS-bearing bacteria and host cells is the critical step in the deployment of the protein secretion machine, yet this process remains poorly understood. Here, we use high-throughput cryo-electron tomography (cryo-ET) to visualize the T3SS-mediated *Salmonella*-host cell interface. Our analysis reveals the intact translocon at an unprecedented level of resolution, its deployment in the host cell membrane, and the establishment of an intimate association between the bacteria and the target cells, which is essential for effector translocation. Our studies provide critical data supporting the long postulated direct injection model for effector translocation.
DOI: https://doi.org/10.7554/eLife.39514.001

*For correspondence:
jliu@yale.edu

Competing interests: The authors declare that no competing interests exist.

## Introduction

Type III secretion systems (T3SSs) are widely utilized by many pathogenic or symbiotic Gram-negative bacteria to directly inject bacterially encoded effector proteins into eukaryotic host cells (*Deng et al., 2017*; *Galán et al., 2014*; *Notti and Stebbins, 2016*). The central element of the T3SS is the injectisome, a multi-protein machine that mediates the selection and translocation of the effectors destined to travel this delivery pathway. The injectisome is highly conserved, both structurally and functionally, across different bacterial species including important pathogens such as *Salmonella*, *Yersinia*, *Shigella*, *Pseudomonas* and *Chlamydia* species. It consists of defined substructures such as the needle complex, the export apparatus, and the cytoplasmic sorting platform (*Galán et al., 2014*; *Hu et al., 2017*; *Loquet et al., 2012*; *Schraidt and Marlovits, 2011*; *Worrall et al., 2016*). The needle complex is composed of a membrane-anchored base, a protruding needle filament, and a tip complex at the distal end of the needle (*Kubori et al., 1998*; *Schraidt et al., 2010*; *Schraidt and Marlovits, 2011*; *Worrall et al., 2016*). The export apparatus, which is formed by several inner membrane proteins, functions as the conduit for substrate

translocation across the bacterial inner membrane (*Dietsche et al., 2016*). The sorting platform is a large cytoplasmic multiple-protein complex that orderly selects and delivers the substrates to the export apparatus (*Lara-Tejero et al., 2011*).

In many bacterial species the activity of these protein injection machines is stimulated upon contact with the target eukaryotic cell membrane, a process thought to be mediated by the tip complex (*Barta et al., 2012*; *Blocker et al., 2008*; *Deane et al., 2006*; *Ménard et al., 1994*; *Zierler and Galán, 1995*). Host cell contact triggers a cascade of poorly understood events that lead to the deployment of the protein translocases onto the host cell membrane where they form a protein channel that mediates the passage of the effector proteins. In the case of the *Salmonella enterica* serovar Typhimurium (*S.* Typhimurium) T3SS encoded within its pathogenicity island 1, the protein translocases are SipB and SipC, which through a process that requires the tip protein SipD, are inserted in the host-cell membrane to form the translocon channel (*Collazo and Galán, 1997*). Deployment of the translocon also results in the intimate association of the bacteria and the host cell, which is orchestrated by the protein translocases themselves (*Lara-Tejero and Galán, 2009*; *Misselwitz et al., 2011*). Despite the critical role of the translocases in intimate attachment and effector translocation, little is known about their structural organization when deployed in the host cell membrane, and previous attempts to visualize them did not provide distinct structural details. This paucity of information is due at least in part to the intrinsic difficulties of imaging bacteria/host cell interactions at high resolution. Here, we used bacterial minicells and cultured mammalian cells combined with high-throughput cryo-ET to study the initial interaction between *S.* Typhimurium and host cells. This experimental system allowed the visualization of the intact translocon deployed in the host cell membrane, in contact with the tip-complex of the T3SS injectisome, at unprecedented resolution. This study provides new insights into the initial events of the T3SS-mediated bacteria-host cell interactions and highlights the potential of cryo-ET as a valuable tool for investigating the host cell-pathogen interface.

## Results

### In situ structures of the T3SS injectisome in the presence or absence of protein translocases

An intrinsic property of many T3SSs is that their activity is stimulated by contact with the target host cell membrane (*Ménard et al., 1994*; *Zierler and Galán, 1995*). This interaction results not only in the stimulation of secretion but also in the deployment of the protein translocases in the host cell membrane, a poorly understood process that is orchestrated by the tip complex of the injectisome's needle filament. In the case of the *S.* Typhimurium SPI-1 T3SS the tip complex is thought to be composed of a single protein, SipD, which organizes as a pentamer at the tip of the needle filament (*Rathinavelan et al., 2014*). However, it has been previously proposed that in *Shigella* spp., in addition to IpaD, a homolog of SipD, the tip complex also contains IpaB, a homolog of SipB (*Cheung et al., 2015*). To get insight into the structural organization of the tip complex prior to bacterial contact with cultured cells, we compared the in situ structures of fully assembled injectisomes from minicells obtained from wild-type, Δ*sipB*, and Δ*sipD S.* Typhimurium strains (*Figure 1A–D*, *Table 1*). We found that injectisomes from wild-type or the Δ*sipB* strains were indistinguishable from one another. In contrast, injectisomes from a Δ*sipD* strain exhibited a shorter needle (~45 nm) in comparison to the needle filaments of injectisomes from the wild-type or Δ*sipB* strains (~50 nm). These observations suggest that SipD is the only structural component of the tip complex (*Figure 1E*). To further explore this hypothesis, we examined by cryo-ET the injectisomes of minicells obtained from *S.* Typhimurium strains expressing FLAG-epitope-tagged versions of SipB, SipC, and SipD, after labeling with anti-FLAG antibodies (*Lara-Tejero and Galán, 2009*) (*Figure 1F–H*, *Figure 1—figure supplement 1*). Only injectisomes from minicells obtained from the strain expressing SipD-FLAG showed the antibodies bound to the needle tip (*Figure 1H*, *Figure 1—figure supplement 1D,I*). This observation is consistent with the notion that, prior to cell contact, SipD is the main, and most likely only component of the tip-complex (*Lara-Tejero and Galán, 2009*).

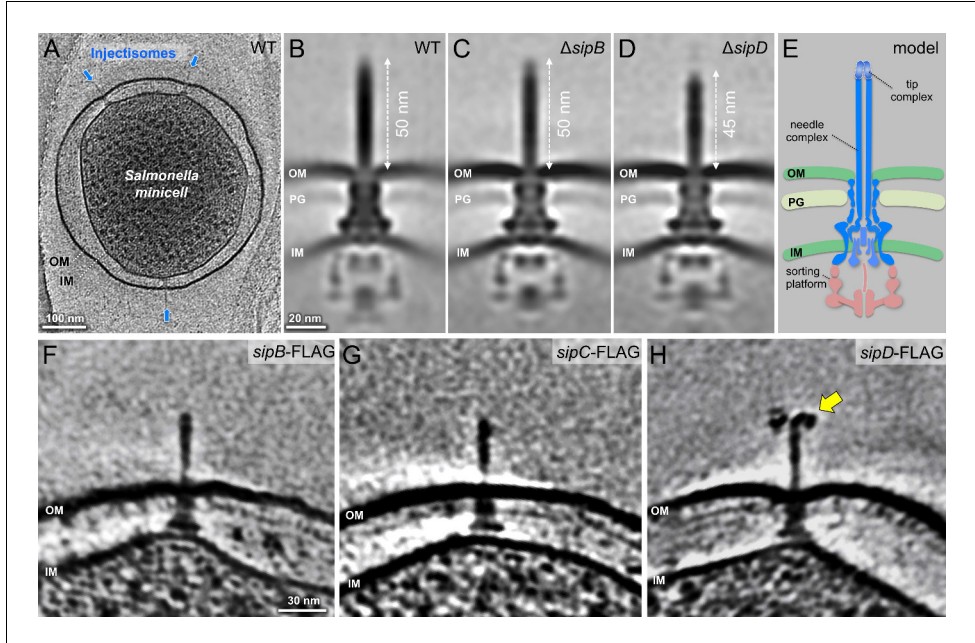

**Figure 1.** In situ structures of host-free *S.* Typhimurium T3SS injectisome in wild-type (WT), Δ*sipB*, and Δ*sipD* minicells. (**A**) A central section of a tomogram showing *S.* Typhimurium minicell containing multiple injectisomes. (**B–D**) Central sections of sub-tomogram averages showing injectisomes of WT, Δ*sipB*, and Δ*sipD*, respectively. (**E**) A schematic of the injectisome. Outer membrane (OM), peptidoglycan (PG), sorting platform, and inner membrane (IM) of *S.* Typhimurium are annotated. (**F–H**) Central sections of tomograms showing injectisomes from strains expressing epitope-tagged (FLAG) SipB, SipC, and SipD, respectively. Yellow arrow indicates antibody bound to the epitope-tag.

DOI: https://doi.org/10.7554/eLife.39514.002

The following figure supplement is available for figure 1:

**Figure supplement 1.** Detection of FLAG-epitope-tagged SipB, SipC, and SipD Central slices from representative tomograms showing.

DOI: https://doi.org/10.7554/eLife.39514.003

## High-resolution imaging of the T3SS mediated *Salmonella*-host cell interface

It is well established that effector translocation through the T3SS requires an intimate association between the bacteria and the host cell (*Grosdent et al., 2002*). It has also been previously demonstrated that such intimate attachment requires an intact type III secretion machine, and in particular, the protein translocases, which most likely mediate such bacteria/host cell interaction (*Lara-Tejero and Galán, 2009*). Despite its central role in effector translocation, however, very little is known about the architecture of this specialized host/bacteria interface. This is largely because of

**Table 1.** Needle lengths of *S.* Typhimurium WT, Δ*sipB*, Δ*sipD* and Δ*sipBCD* cells.
A summary of statistical measures including needle length average, standard deviation, and standard error of mean. Data were compared using an unpaired *t* test.

|  | Sample size | Average (nm) | Standard Deviation | Standard Error of Mean(nm) | *P* value campared to WT |
|---|---|---|---|---|---|
| WT | 135 | 51.0 | 4.8 | 0.42 |  |
| Δ*sipB* | 46 | 50.6 | 4.0 | 0.59 | 0.62 |
| Δ*sipD* | 61 | 46.5 | 3.9 | 0.50 | <0.0001 |
| Δ*sipBCD* | 46 | 45.3 | 3.0 | 0.44 | <0.0001 |

DOI: https://doi.org/10.7554/eLife.39514.004

the lack of amenable experimental approaches that would allow a detail view of this interface. Cryo-ET is uniquely suited to examine host/pathogen interactions at high resolution. However, sample thickness limits the utility of this approach. To get around this limitation we used bacterial minicells as a surrogate for whole bacteria since it has been previously shown that they are capable of assembling functional T3SS injectisomes that can deliver de novo synthesized T3SS substrates into cultured cells (*Carleton et al., 2013*). However, minicells are inefficient at triggering membrane ruffling, actin filament remodeling, and bacterial internalization due to inefficient partitioning of the effector proteins that trigger these responses. Consequently, while minicells are proficient at establishing a T3SS-mediated intimate association with cultured epithelial cells, they are inefficient at triggering their own internalization thus remaining firmly attached on the cell surface. These features make them ideally suited for high-resolution cryo-ET imaging. Therefore, we applied bacterial minicells obtained from wild-type *S.* Typhimurium onto cultured epithelial cells grown on cryo-EM grids. We found that the periphery of adherent cells is sufficiently thin (<500 nm) for high-resolution imaging (*Figure 2—figure supplement 1*). We readily observed T3SS injectisomes at the interface between minicells and the plasma membrane of cultured epithelial cells (*Figure 2A,B*). We found that in the presence of the injectisomes, the spacing between the surface of the *S.* Typhimurium minicells and the cultured-cell plasma membrane was ~50 nm, which matches the needle length of the injectisome imaged prior to their application to cultured cells (*Figure 2—figure supplement 2A–F,M*). The orientation of the injectisomes in the bacteria/target cell interface was perpendicular relative to the host PM, and the needle of the host-interacting injectisomes appeared straight (*Figure 2C*). We also observed that the interaction of the injectisome and the target cell resulted in a noticeable inward bend of the PM (*Figure 2C*, *Video 1*). Consistent with this observation, the distance between the bacterial cell and the PM was shorter (~30 nm) than the distance observed in areas immediately adjacent to the injectisomes (*Figure 2—figure supplement 2A–L*). However, we did not observe any sign of penetration of the needle filament through the host cell plasma membrane as it has been previously proposed (*Hoiczyk and Blobel, 2001*). The length of the bacterial-envelope-embedded injectisome base substructure before (30.5 ± 2.3 nm) and after (30.8 ± 2.2 nm) the bacteria/target cell interactions remained unchanged (*Figure 2—figure supplement 2M,N*). This is in contrast to the *Chlamydia* T3SS, which has been reported to undergo significant conformational changes upon contact with host cells (*Nans et al., 2015*). The reasons for these differences are unclear and may either reflect intrinsic differences between these T3SS, or differences in the methodology used, which resulted in higher resolution of the visualized *S.* Typhimurium T3SS structures. Together, these observations indicate that (1) the interactions of the T3SS injectisome with the target cell results in the bending of the PM without penetration of the needle filament, and (2) upon contact with target cells the injectisome does not undergo conformational changes that could be seen at this level of resolution.

## Visualization of the formation of the translocon in the target host cell membrane

The deployment of the translocon is an essential step in the T3SS-mediated delivery of effector proteins. However, very little information is available on both, the architecture of the assembled translocon, as well as the mechanisms leading to its deployment on the target cell. It is believed that the deployment process must be initiated by a sensing step most likely mediated by the tip complex (i.e. SipD), a step that must be followed by the subsequent secretion of the translocon components (i.e. SipB and SipC) destined to be inserted on the target eukaryotic cell PM. To capture the formation of the translocon, we analyzed over 600 injectisomes adjacent to the host PM. Classification of sub-tomograms depicting the region of the tip complex (*Figure 3A*) showed the PM at various conformations and distances to the needle tip (*Figure 3B–I*), which presumably represent intermediate steps prior to the deployment of the translocon and the resulting intimate attachment of the bacteria to the PM. After further alignment and classification of the injectisomes in intimate association with the PM, we obtained a distinct structure of the putative translocon in the host PM (*Figure 3J*). Sub-tomogram averages of injectisomes from the *S.* Typhimurium translocase-deficient mutants ΔsipB or ΔsipD in close proximity to the target cell PM did not show this distinct structure (*Figure 3K,L*), thus confirming that this density most likely corresponds to the assembled translocon. To better visualize the translocon in 3D, we segmented the distinct translocon structure in the context of the host PM, the needle, and its tip complex (*Figure 3M,N*). We found that the translocon

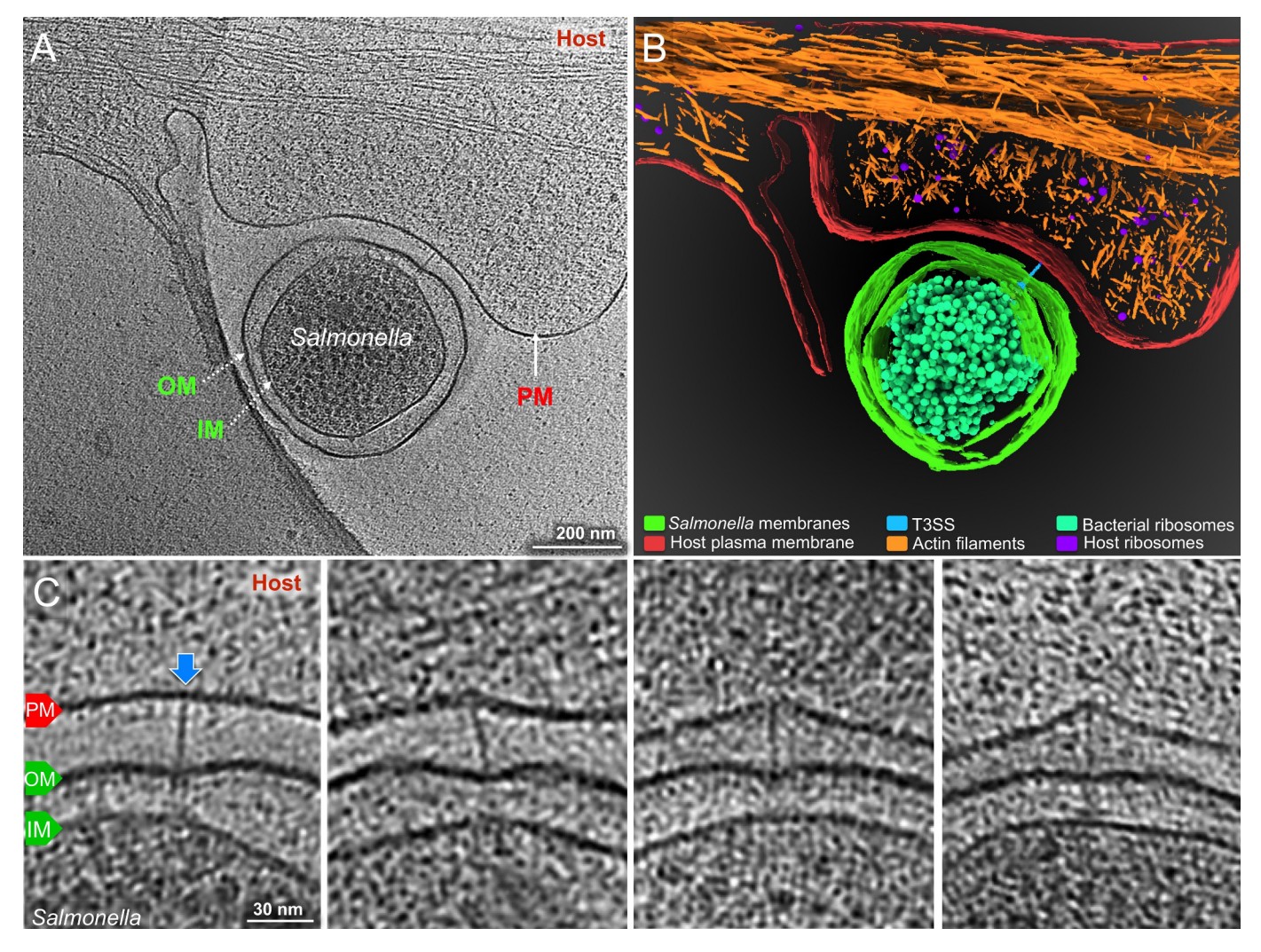

**Figure 2.** Visualization of the T3SS mediated *Salmonella*-Host interactions. (**A**) A central slice showing a *S.* Typhimurium minicell interacting with a host. Plasma membrane (PM) of HeLa cell, outer membrane (OM) and inner membrane (IM) of *S.* Typhimurium are annotated. (**B**) 3D rendering of the tomogram shown in (**A**). (**C**) Tomographic slices showing injectisomes interacting with the host PM. Blue arrows indicate needles attached to the host PM. Direction of the arrow represents the angle of needle perpendicular to the host PM.

DOI: https://doi.org/10.7554/eLife.39514.005

The following figure supplements are available for figure 2:

**Figure supplement 1.** Cultivation of mammalian cells (HeLa) on EM grid for cryo-ET.
DOI: https://doi.org/10.7554/eLife.39514.006

**Figure supplement 2.** Inter-membrane spacing between outer membrane and plasma membrane.
DOI: https://doi.org/10.7554/eLife.39514.007

has a thickness of 8.1 nm spanning the host PM and a diameter of 13.5 nm on its protruding portion (*Figure 3J*). This size is substantially smaller than reported size of the translocon of enteropathogenic *E. coli* assembled from purified proteins in vitro, which was estimated to be 55–65 nm in diameter (*Ide et al., 2001*). One half of the translocon is embedded in the host PM, while the other half protrudes towards the host cytoplasm. In the middle of the protruded portion, we observed a hemispherical hole, which may represent the channel through which effectors make their way into the target cell plasma membrane (*Figure 3N*). The presence of this structure is entirely consistent with the long-standing notion that the translocon forms a conduit through the host PM to facilitate the translocation of effectors (*Mueller et al., 2008*).

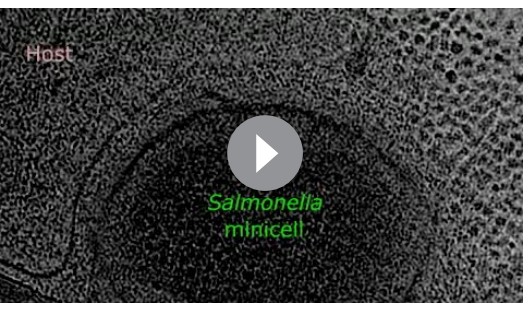

**Video 1.** A typical reconstruction shows the detailed interaction between the T3SS machines and the target cell and the membrane remodeling.
DOI: https://doi.org/10.7554/eLife.39514.008

Comparison of the arrangement of the injectisomes in relation to the target cell PM in wild-type and translocase-deficient strains revealed marked differences. In comparison to wild-type, bacterial cells obtained from translocase-deficient mutants showed a smaller proportion of injectisomes attached to the host PM (*Figure 4A*). We also noticed that, unlike wild-type injectisomes, which most often appeared perpendicular to the target cell PM (*Figure 2C*), the injectisomes from the translocase deficient mutant strains Δ*sipB*, Δ*sipD*, or Δ*sipBCD* appeared arranged at various angles relative to the PM (*Figure 4B–J*, *Figure 4—figure supplement 1*). These observations are consistent with the observations indicating that in the absence of the translocases, the injectisomes do not intimately attach to the target cell PM (*Lara-Tejero and Galán, 2009*). These data also further support the notion that the distinct structure embedded in host membrane in close apposition to the T3SS injectisome needle tip is formed by the translocon.

One of the striking features associated with the intimate T3SS mediated contact and the formation of the translocon is the target cell PM remodeling around the translocon-injectisome needle tip interface, appearing in a 'tent-like' conformation (*Figure 2C*, *Video 1*). This feature is likely the result of the close association between the bacteria and the target cell presumably mediated not only by the T3SS but also by multiple additional adhesins encoded by *S.* Typhimurium. In fact, the distance of the bacteria OM and the target cell is shorter than the length of the needle itself, which results in the bending of the target cell PM and the 'tent-like' conformation around the injectisome target cell PM interface. It is possible that this intimate association may facilitate the T3SS-mediated translocation of effector proteins (*Figure 5*, *Video 2*). These observations are also consistent with previous reports indicating that needle length, which presumably influences the ability of the bacteria to 'push' the needle against the host cell membrane, does contribute to type III secretion translocation efficiency (*Mota et al., 2005*).

## Discussion

We have presented here a high-resolution view of the interface between the *S.* Typhimurium T3SS injectisome and the target eukaryotic cell plasma membrane, which has provided new details on the intimate attachment of this pathogen that precedes T3SS-mediated effector protein translocation. Notably, we observed a well-defined 'bend' on the target cell membrane in areas of the bacteria/host cell membrane interface surrounding the needle filament. These observations reflect the intimate attachment that is known to be required for optimal T3SS-mediated effector translocation. Importantly, we have been able to visualize a distinct density within the region of the target host cell membrane in close apposition to the needle tip of the T3SS injectisome. We present evidence that this density corresponds to the deployed T3SS translocon since this density was absent in the bacteria/PM interface of mutant bacteria that lack the translocon components. It has been previously shown that protein translocases SipB and SipC form a complex in the host membrane (*Hayward and Koronakis, 1999*; *Myeni et al., 2013*). Single-molecule fluorescence photobleaching experiments in a *Pseudomonas aeruginosa* T3SS further suggested that two translocases form a hetero-complex of defined stoichiometry in a membrane bilayer (*Romano et al., 2016*). Although our in situ structure of the translocon is not sufficient in detail to determine its stoichiometry, it appears to be large enough to accommodate multiple copies of SipB/SipC complex. The translocon is partially embedded in the host membrane. A large portion of the translocon extends towards the host cytoplasm, which is consistent with results in *P. aeruginosa* showing that the translocase PopB is stably inserted into lipid bilayers with its N-terminal domain and C-terminal ends exposed to the host cytoplasm (*Discola et al., 2014*). The dimensions of the in situ translocon structure determined in this study (~13.5 nm in diameter, 8 nm in thickness) are much smaller than previous estimates (50–65 nm in

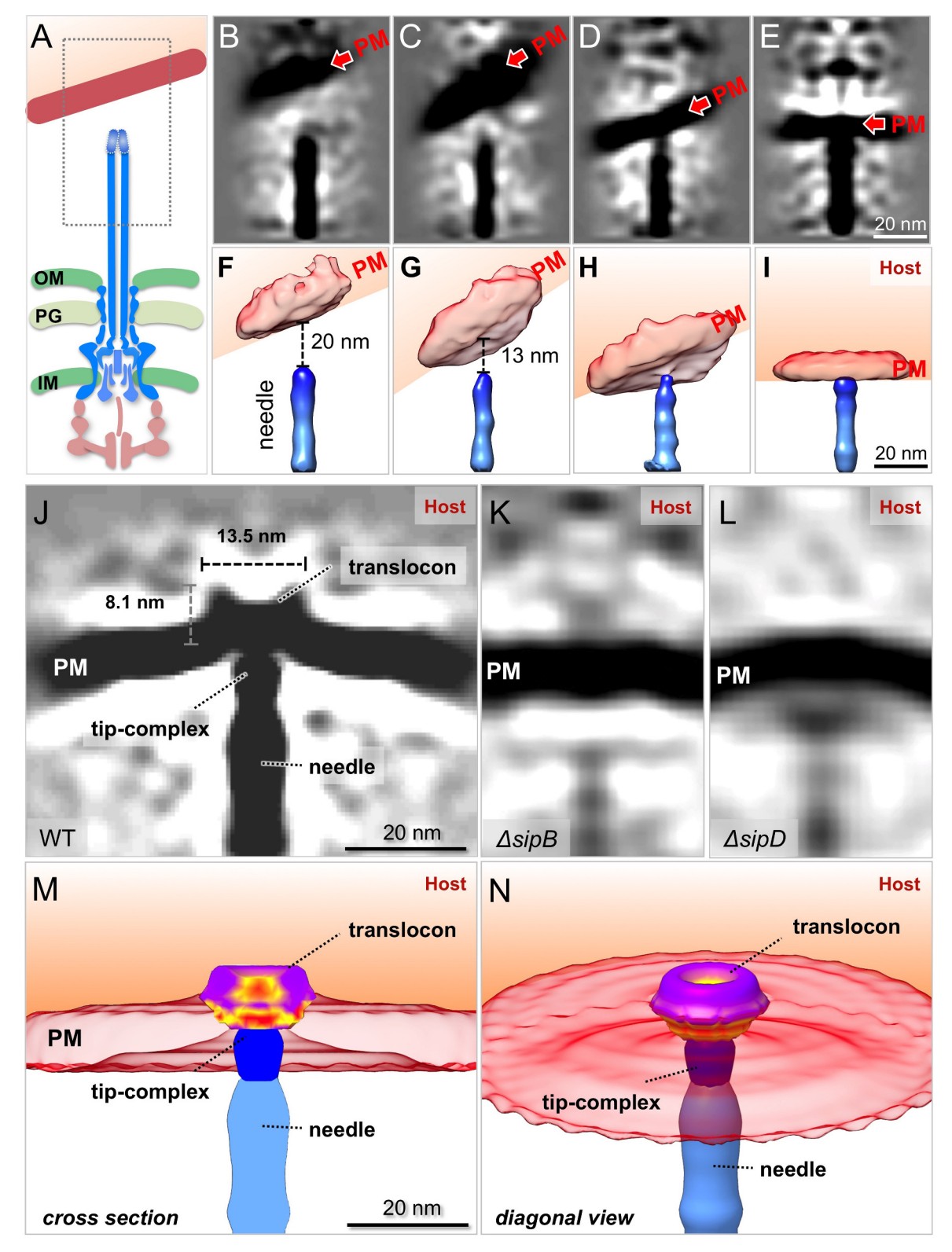

**Figure 3.** In situ structural analysis of the interface between the T3SS needle and the host membrane reveals a novel structure of the intact translocon. (A) A schematic representation of the *S.* Typhimurium injectisome with a box highlighting the area used for alignment and classification. (B–E) Central sections and (F–I) 3-D surface views of class averages showing different spacings between the needle and the plasma membrane (PM). (J–L) Central

*Figure 3 continued on next page*

*Figure 3 continued*

sections of the sub-tomogram averages of the interface between the host PM and the needle of WT, Δ*sipB*, and Δ*sipD*, respectively. (M) Cross-section and (N) diagonal view of the surface rendering of the translocon in panel (J).

DOI: https://doi.org/10.7554/eLife.39514.009

diameter) obtained from the observation of EPEC translocons assembled from purified components on red blood cells (*Ide et al., 2001*). It is unlikely that these differences may reflect substantial differences between the dimensions of translocons from different T3SSs. It is possible that the observed differences may reflect differences in the experimental approaches used in the different studies. However, most likely these observations indicate fundamental differences in the translocon assembly mechanisms from purified components in comparison to translocon assembly during bacteria/target cell membrane interactions. It is well established that the deployment of the translocon during

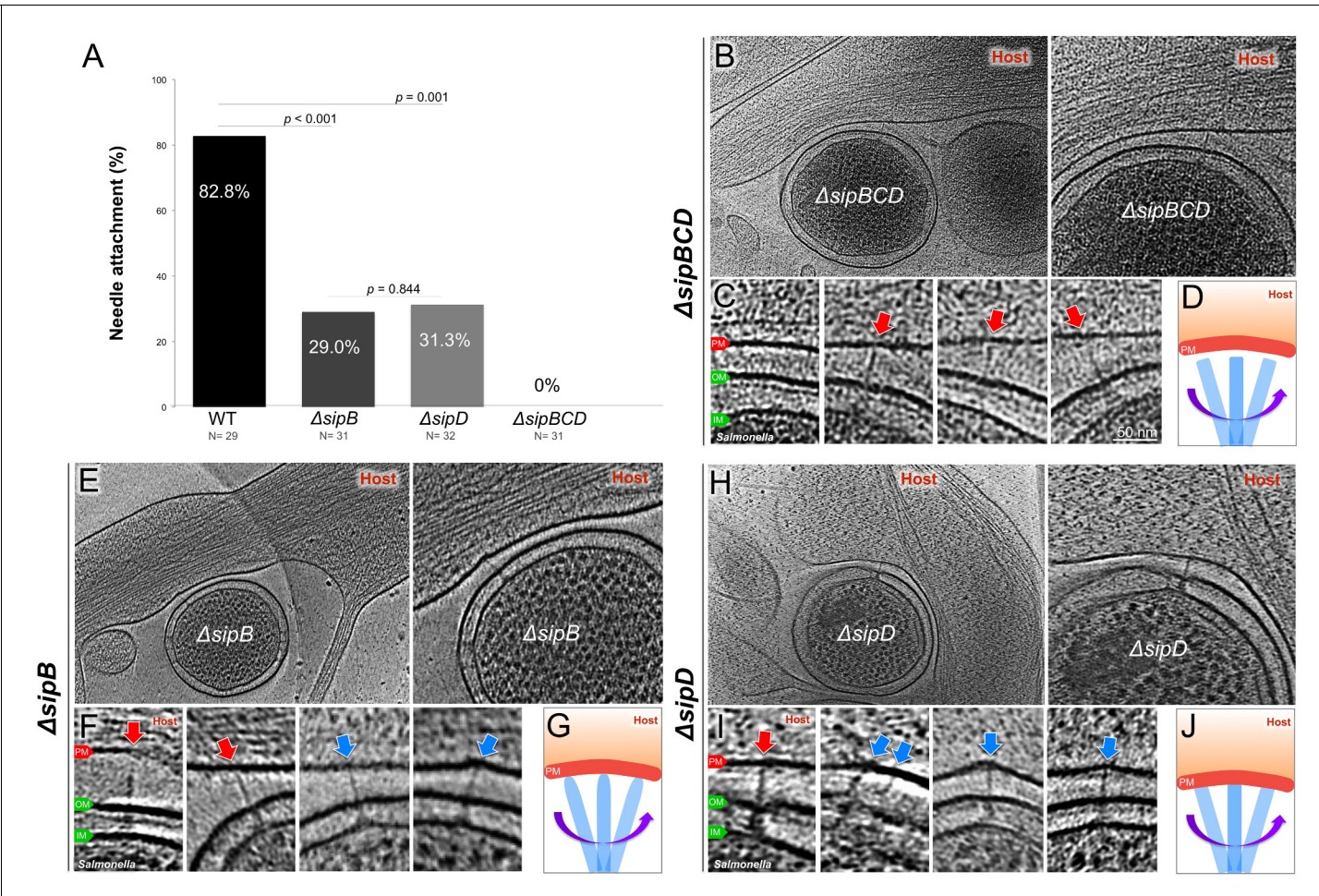

**Figure 4.** Deletion of the protein translocases disrupts the T3SS-dependent intimate attachment to the host PM, and the formation of the translocon. (A) Percentage of minicells attached to the host membrane via needle-membrane contact. Data were compared using a chi-squared test. (B, C) Central slices from tomograms showing the Δ*sipBCD* injectisomes interacting with the host PM. (E, F) Central slices from tomograms showing the Δ*sipB* injectisomes interacting with the host PM. (H, I) Central slices from tomograms showing the Δ*sipD* injectisomes interacting with the host PM. Blue arrows indicate needles attached to the host PM. Red arrows indicate unattached needles. (D, G, J) Schematic models depicting needle-attachment patterns from three mutants, respectively.

DOI: https://doi.org/10.7554/eLife.39514.010

The following figure supplement is available for figure 4:

**Figure supplement 1.** Gallery of snapshots from host-free and host-interacting *S.* Typhimurium minicells.

DOI: https://doi.org/10.7554/eLife.39514.011

**Figure 5.** Model of the *S.* Typhimurium injectisome interacting with the host cell membrane. (**A**) A schematic diagram of *S.* Typhimurium interacting with the host cell. (**B**) Molecular model of the T3SS injectisome at the *Salmonella*-host cell interface.

DOI: https://doi.org/10.7554/eLife.39514.012

bacterial infections is orchestrated by the tip complex of the T3SS injectisome. In the absence of the tip protein, the components of the translocon are very efficiently secreted but they are unable to form the translocon (*Kaniga et al., 1995*; *Ménard et al., 1994*). It is therefore possible that the insertion in the membrane of the purified translocon components in the absence of the tip protein may lead to a structure that is substantially different from the one that results from the interaction of bacteria with target cells.

Contrary to what has been previously proposed for the Chlamydia T3SS (*Nans et al., 2015*), we did not observe any obvious conformational changes in the injectisomes prior and post interaction with host cells. It is unlikely that these observations are an indication of fundamental differences between the T3SS injectisomes in different bacteria. Rather, the differences observed might reflect differences in the experimental approaches used in our studies, which resulted in a substantially higher resolution.

In summary, our studies have provided a close-up view of the interface between the T3SS injectisome and the target cell PM, which has resulted in the visualization of the deployed T3SS translocon complex. Importantly, given the highly conserved nature of the T3SSs among many Gram-negative bacteria, our studies have broad

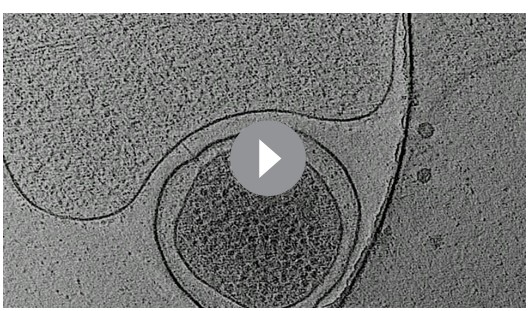

**Video 2.** An animation shows the T3SS mediated *Salmonella*-host interaction and a plausible pathway of effector translocation.

DOI: https://doi.org/10.7554/eLife.39514.013

scientific implications and provide a paradigm for the study of host-pathogen interactions in a greater detail.

# Materials and methods

## Key resources table

| Reagent type (species) or resources | Designation | Source or reference | Identifiers | Additional information |
|---|---|---|---|---|
| Strain, strain background | SB1780 (*Salmonella enterica* serovar Typhymurium SL1344) | PMID: 23481398 | *minD::cat (wt)* | Galán Laboratory (Yale University) |
| Strain, strain background | SB3542 | This study | Δ*sipB minD::cat* | Galán Laboratory (Yale University) |
| Strain, strain background | SB3543 | This study | Δ*sipD minD::cat* | Galán Laboratory (Yale University) |
| Strain, strain background | SB3141 | This study | Δ*sipBCD minD::cat* | Galán Laboratory (Yale University) |
| Strain, strain background | SB3046 | PMID: 28283062 | Δ*spaO minD::cat* | Galán Laboratory (Yale University) |
| Strain, strain background | SB3544 | This study | *sipB3xFLAG minD::cat* | Galán Laboratory (Yale University) |
| Strain, strain background | SB3545 | This study | *sipB3xFLAG minD::cat* | Galán Laboratory (Yale University) |
| Strain, strain background | SB3546 | This study | *sipB3xFLAG minD::cat* | Galán Laboratory (Yale University) |
| Genetic reagent | pSB3292 (Plasmid) | PMID: 28283062 | *hilA* in pBAD24 | Galán Laboratory (Yale University) |
| Genetic reagent | *minD::cat* P22 (P22 bacteriophage lysate) | Galán Laboratory (Yale University) | P22 lysate from SB1780 *S.* Typhimurium strain | Source of *minD::cat* allele |
| Cell line | HeLa | ATCC | Hela (ATCC CCL-2) | |
| Antibody | M2 | Sigma-Aldrich | F3165 | 1:1000 by volume |
| Chemical compound, drug | LB Broth | Fisher BioReagents | BP1426 | |
| Chemical compound, drug | LB Agar | Fisher BioReagents | BP1425 | |
| Chemical compound, drug | L-arabinose | VWR | 1B1473 | |
| Chemical compound, drug | Ampicillin Sodium Salt | Fisher BioReagents | BP1760-25 | |
| Other | Gold grid | Quantifoil | R 2/1 on Au 200 mesh | |
| Software, algorithm | SerialEM | PMID: 16182563 | http://bio3d.colorado.edu/SerialEM/ | Data acquisition |
| Software, algorithm | MotionCor2 | PMID: 28250466 | http://msg.ucsf.edu/em/software/motioncor2.html | Motion correction |
| Software, algorithm | Tomoauto | PMID: 26863591 | https://github.com/DustinMorado/tomoauto | Tomogram reconstruction |
| Software, algorithm | Tomo3D | PMID: 25528570 | https://sites.google.com/site/3demimageprocessing/tomo3d | Tomogram reconstruction |
| Software, algorithm | IMOD | PMID: 8742726 | http://bio3d.colorado.edu/imod/ | Tomogram reconstruction |
| Software, algorithm | I3 | PMID: 16973379 | http://www.electrontomography.org/ | Sub-tomogram averaging |

*Continued on next page*

*Continued*

| Reagent type (species) or resources | Designation | Source or reference | Identifiers | Additional information |
|---|---|---|---|---|
| Software, algorithm | UCSF Chimera | PMID: 15264254 | http://www.cgl.ucsf.edu/chimera/ | 3D rendering |
| Software, algorithm | UCSF ChimeraX | PMID: 28710774 | https://www.rbvi.ucsf.edu/chimerax/ | 3D rendering |

## Bacterial strains

The minicell producing *S.Typhimurium ΔminD*, which is referred to in this study as wild-type, has been previously described (*Carleton et al., 2013*; *Hu et al., 2017*). Mutations in the genes encoding the translocases (*ΔsipB*, *ΔsipC*) or tip complex (*ΔsipD*) proteins where introduced in into the *ΔminD S.* Typhimurium strain by allelic exchange as previously described (*Lara-Tejero et al., 2011*). The strains were listed in Key Resources Table.

## Isolation of minicells

Minicell producing bacterial strains were grown overnight at 37°C in LB containing 0.3M NaCl. Fresh cultures were prepared from a 1:100 dilution of the overnight culture and then grown at 37°C to late log phase in the presence of ampicillin (200 µg/mL) and L-arabinose (0.1%) to induce the expression of regulatory protein HilA and thus increase the number of injectisomes partitioning to the minicells (*Carleton et al., 2013*). To enrich for minicells, the culture was centrifuged at 1000 x g for 5 min to remove bacterial cells, and the supernatant fraction was further centrifuged at 20,000 x g for 20 min to collect the minicells. The minicell pellet was resuspended in Dulbecco's Modified Eagles Medium (DMEM) prior to their application to cultured HeLa cells.

## Antibody labeling

Minicells expressing 3xFLAG-epitope-tagged versions of SipB, SipC, and SipD were incubated with a saturating amount of anti-FLAG antibody (1:1000 by volume) for 30 min at room temperature. After incubation, minicells were pelleted and resuspended in a fresh LB broth containing ampicillin (200 µg/ml) for 3 times to remove unbound antibodies.

**Table 2.** Number of tomograms collected and analyzed.

| | Number of tomograms collected |
|---|---|
| WT - HeLa | 458 |
| ΔsipB - HeLa | 46 |
| ΔsipD - HeLa | 52 |
| ΔsipBCD - HeLa | 86 |
| ΔspaO - HeLa | 84 |
| WT minicells | 85 |
| ΔsipB minicell | 115 |
| ΔsipD minicell | 142 |
| sipB-FLAG - HeLa | 9 |
| sipC-FLAG - HeLa | 8 |
| sipD-FLAG - HeLa | 11 |
| sipB-FLAG | 5 |
| sipB-FLAG | 7 |
| sipB-FLAG | 13 |
| Total | 1051 |

DOI: https://doi.org/10.7554/eLife.39514.014

## HeLa cell culture on EM grid and infection

HeLa cells were cultured in DMEM supplemented with 10% fetal bovine serum and gentamicin (50 µg/ml). The day before plating, gold EM grids with 2/1 Quantifoil were placed in glass bottom Mat-Tek dishes (facilitating fluorescence imaging and removal for cryo-preservation) and coated with 0.1 mg/ml poly-D-lysine overnight at 37°C. After rinsing the grids with sterile water, freshly trypsinized HeLa cells were plated on top of the pre-treated grids that were allowed to grow overnight at 37°C/ 5% $CO^2$. To infect HeLa cells with *S.* Typhimurium minicells, grids with adherent HeLa cells were removed from the culture dish and minicells were directly applied to the grids.

## Vitrification and cryoEM sample preparation

At different time points after infection, the EM grids with HeLa cells and *S.* Typhimurium minicells were blotted with filter paper and vitrified in liquid ethane using a gravity-driven plunger apparatus as described (*Hu et al., 2017*; *Hu et al., 2015*).

## Cryo-ET data collection and reconstruction

The frozen-hydrated specimens were imaged with 300kV electron microscopes. 713 tomograms were acquired from single-axis tilt series at ~6 µm defocus with cumulative does of ~80 e⁻/Å² using Polara equipped with a field emission gun and a direct detection device (Gatan K2 Summit). 313 tomograms were acquired from single-axis tilt series at ~1 µm defocus with cumulative does of ~50 e⁻/Å² using Titan Krios equipped with a field emission gun, an energy filter, Volta phase plate, and a direct detection device (Gatan K2 Summit). The tomographic package SerialEM (*Mastronarde, 2005*) was utilized to collect 35 image stacks at a range of tilt angles between −51° and +51° for each data set. Each stack contained 10–15 images, which were first aligned using Motioncorr (*Li et al., 2013*) and were then assembled into the drift-corrected stacks by TOMOAUTO (*Hu et al., 2015*). The drift-corrected stacks were aligned and reconstructed by using marker-free alignment (*Winkler and Taylor, 2006*) or IMOD marker-dependent alignment (*Kremer et al., 1996*). In total, 1051 tomograms (3,600 × 3,600 × 400 pixels) were generated for detailed examination of the *Salmonella*-host interactions (*Table 2*). The softwares used in the study were listed in Key Resources Table.

## Sub-tomogram analysis

Sub-tomogram analysis was accomplished as described previously (*Hu et al., 2015*) to analyze over 700 injectisomes extracted from 458 tomograms. Briefly, we first identified the injectisomes visually on each minicell. Two coordinates along the needle were used to estimate the initial orientation of each particle assembly. For initial analysis, 4 × 4 × 4 binned sub-tomograms (128 × 128 × 128 voxels) of the intact injectisome were used for alignment and averaging by using the tomographic package I3 (*Winkler and Taylor, 2006*; *Winkler et al., 2009*). Then multivariate statistical analysis and hierarchical ascendant classification were used to analyze the needle tip complex (*Winkler et al., 2009*).

## 3-D visualization and molecular modeling

Outer membrane (OM) and inner membrane (IM) of *S.* Typhimurium, Plasma membrane (PM) of HeLa cells, actin filaments, and ribosomes were segmented using EMAN2 (*Chen et al., 2017*). UCSF Chimera (*Pettersen et al., 2004*) and UCSF ChimeraX (*Goddard et al., 2018*) were used to visualize the sub-tomogram average structures in 3-D and build atomic model of the T3SS injectisome. The atomic model was built as described briefly (*Hu et al., 2017*) except for the basal body, which we docked PDB-5TCR (*Worrall et al., 2016*) and PDB-3J1W (*Bergeron et al., 2013*). Video clips for the supplemental videos were generated using UCSF Chimera, UCSF Chimera X, and IMOD, and edited with iMovie.

## Distance measurement and statistical analysis

IMOD (3dmod Graph) was used to measure lengths (in pixels) of various features. Each measurement was recorded in MS Excel for statistical analysis: Mean, standard deviation, standard error of mean, and Welch's t-test.

## Acknowledgements

This work was supported by Grants AI030492 (to JEG) from the National Institute of Allergy and Infectious Diseases, and GM107629 from the National Institute of General Medicine (to JL).

## Additional information

### Funding

| Funder | Grant reference number | Author |
|---|---|---|
| National Institute of Allergy and Infectious Diseases | AI030492 | Jorge E Galán |
| National Institute of General Medical Sciences | GM107629 | Jun Liu |

The funders had no role in study design, data collection and interpretation, or the decision to submit the work for publication.

### Author contributions

Donghyun Park, Data curation, Formal analysis, Validation, Investigation, Visualization, Writing—original draft; Maria Lara-Tejero, Data curation, Investigation, Methodology, Writing—original draft; M Neal Waxham, Wenwei Li, Bo Hu, Data curation, Investigation; Jorge E Galán, Conceptualization, Supervision, Investigation, Writing—original draft, Project administration, Writing—review and editing; Jun Liu, Conceptualization, Supervision, Investigation, Methodology, Writing—original draft, Project administration, Writing—review and editing

### Author ORCIDs

Donghyun Park ![ORCID] http://orcid.org/0000-0003-2048-6004
Jorge E Galán ![ORCID] http://orcid.org/0000-0002-6531-0355
Jun Liu ![ORCID] http://orcid.org/0000-0003-3108-6735

### Decision letter and Author response

Decision letter https://doi.org/10.7554/eLife.39514.017
Author response https://doi.org/10.7554/eLife.39514.018

## Additional files

### Supplementary files

• Transparent reporting form
DOI: https://doi.org/10.7554/eLife.39514.015

### Data availability

All data generated or analysed during this study are included in the manuscript and supporting files.

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
