## [Decision Letter]

Thank you for submitting your article "Visualization of the type III secretion mediated *Salmonella*-host cell interface using cryo-electron tomography" for consideration by *eLife*. Your article has been reviewed by three peer reviewers, and the evaluation has been overseen by a Reviewing Editor and Andrea Musacchio as the Senior Editor. The following individuals involved in review of your submission have agreed to reveal their identity: Eric Cascales (Reviewer #1); Samuel Wagner (Reviewer #3).

The reviewers have discussed the reviews with one another and the Reviewing Editor has drafted this decision to help you prepare a revised submission.

Summary:

Based on all of the three reviews, the work can be described with the reviewer's assessment that this "is a beautiful new look into the biology of the T3SS (Type 3 Secretion System) employing cutting edge methods to get at decades-old, unsolved questions in how this critical virulence system functions."

Essential revisions:

However, the following concerns brought up by reviewers are valid:

1) It is not entirely convincing to use the antibody-based approach to label SipB, SipC and SipD. The extra-density shown at the needle tip in Figure 1H could correspond to anything. Additional data should be provided to support the claim that this extra-density corresponds to an antibody specifically bound to the tip (and not random background). May be an anti-FLAG antibody coupled to anything (additional enzymatic domain, gold particle, etc.) will be sufficient.

2) Similarly, there is no proof that the extra-density in Figure 3J corresponds to the translocon rather than host cell membrane rearrangements upon needle attachment. Here again, supplementary data should be provided to unambiguously support the author's statement.

3) The study is lacking rigorous statistical analyses and significance. For most of the experiments, the sample size, the standard deviation and the statistical significance are missing. These should be added for the final submission.

4) Some discussion of previous work relating to the transmembrane topology and stoichiometry of translocator proteins are missing.

- Translocon structure with cytoplasmic protrusion suggests that most of the translocon protein is on the cytoplasmic side of the membrane. This is in contrast to previous studies on the topologies of the *Pseudomonas* homologs of these proteins. See Discola et al., 2014.

- Please reflect on the finding here and the stoichiometries reported for in vitro reconstituted *Pseudomonas* translocons: Romano et al., 2016.

- Furthermore, the "tent-like" PM around the translocon suggests that considerable tension is required for proper filament-tip-translocon interaction and injection of proteins. Please discuss this in the light of the "Needle length does matter" findings by the Cornelis group. Mota et al., 2005.

---

## [Author Response]

1) It is not entirely convincing to use the antibody-based approach to label SipB, SipC and SipD. The extra-density shown at the needle tip in Figure 1H could correspond to anything. Additional data should be provided to support the claim that this extra-density corresponds to an antibody specifically bound to the tip (and not random background). May be an anti-FLAG antibody coupled to anything (additional enzymatic domain, gold particle, etc.) will be sufficient.

We appreciate the comment and agree that images we had provided for this experiment were somewhat ambiguous. To provide better evidence for our findings, we have re-conducted the same experiment with minor optimizations in the imaging condition and sample preparation. Our new data showed convincingly the presence of well-defined density for the antibodies at the needle tip on both raw tomograms and sub-tomogram averages. Therefore, we have replaced Figures 1F-Hwith these new results. In addition, we have added Figure 1—figure supplement 1 to support our findings with more raw tomographic slices, sub-tomogram averages and detailed quantification.

2) Similarly, there is no proof that the extra-density in Figure 3J corresponds to the translocon rather than host cell membrane rearrangements upon needle attachment. Here again, supplementary data should be provided to unambiguously support the author's statement.

We appreciate the comment. Although we have not provided direct proof that the extra-density we observed corresponds to the translocon, a technical impossibility at this juncture, we have provided what we believe to be strong indirect evidence that the extra-density indeed corresponds to the translocon. We have for example shown that the density corresponding to the translocon is absent in two translocon defective mutants (Figure 3K, L). Importantly, the extra-density is specifically observed at the top of the tip complex, which is formed by SipD. The translocon is indeed the only known protein complex that is assembled in the host membrane at this location. Furthermore, the structural change resulted from the formation of the translocon (Figure 3J) is strikingly different from the membrane rearrangement resulted from the needle attachment (Figure 2C and Figure 4F, I). Therefore, we believe that our data provide strong evidence that the extra-density is formed by the translocon.

3) The study is lacking rigorous statistical analyses and significance. For most of the experiments, the sample size, the standard deviation and the statistical significance are missing. These should be added for the final submission.

We thank the reviewers for bringing up this point. We have added appropriate statistical measures to the revised Figure 4A, Figure 1 —figure supplement 1I, Figure 2—figure supplement 2M, N, and Table 1, 2. As expected, all the significance values are in good agreement with our previous data interpretations.

4) Some discussion of previous work relating to the transmembrane topology and stoichiometry of translocator proteins are missing.- Translocon structure with cytoplasmic protrusion suggests that most of the translocon protein is on the cytoplasmic side of the membrane. This is in contrast to previous studies on the topologies of the Pseudomonas homologs of these proteins. See Discola et al., 2014.

*- Please reflect on the finding here and the stoichiometries reported for* in vitro *reconstituted Pseudomonas translocons: Romano et al., 2016.*

- Furthermore, the "tent-like" PM around the translocon suggests that considerable tension is required for proper filament-tip-translocon interaction and injection of proteins. Please discuss this in the light of the "Needle length does matter" findings by the Cornelis group. Mota et al., 2005.

We thank the reviewers for suggesting a list of papers that, indeed, are highly relevant to our manuscript.

Discola et al.: As mentioned in our previous Discussions section, we believe that the different topology of in vitro assembled *Pseudomonas* translocon suggested by Discola et al. may reflect differences in experimental approaches. We have discussed and cited this paper in the Discussion.

Romano et al.: We have considered discussing the issue of the putative stoichiometry of the in-vitro assembled complex. However, our current resolution of the translocon is insufficient to credibly discuss issues related to the translocon stoichiometry. In addition, to our knowledge no stoichiometry data on in-vivo assembled translocon is available and it is not completely clear that the in-vitro assembled translocon accurately reflects the organization of the in vivo assembled structure. More studies at higher resolution will be required to address this issue in the future.

Mota et al.: We believe that the findings in Mota et al. does indeed provide support to our “tent-like” conformation model of the host membrane rearrangement. Therefore we have discussed and cited this paper in our revised Discussion as suggested by the reviewers.